# Improving Atorvastatin Release from Polyelectrolyte Complex-Based Hydrogels Using Freeze-Drying: Formulation and Pharmaceutical Assessment of a Novel Delivery System for Oral Candidiasis Treatment

**DOI:** 10.3390/ijms26052267

**Published:** 2025-03-04

**Authors:** Joanna Potaś-Stobiecka, Radosław Aleksander Wach, Bożena Rokita, Weronika Kaja Simonik, Magdalena Wróblewska, Karolina Borkowska, Silje Mork, Nataša Škalko-Basnet, Katarzyna Winnicka

**Affiliations:** 1Department of Pharmaceutical Technology, Faculty of Pharmacy, Medical University of Białystok, Mickiewicza 2C, 15-222 Białystok, Poland; magdalena.wroblewska@umb.edu.pl (M.W.); katarzyna.winnicka@umb.edu.pl (K.W.); 2Institute of Applied Radiation Chemistry, Faculty of Chemistry, Łódź University of Technology, Wróblewskiego 15, 93-590 Łódź, Poland; radoslaw.wach@p.lodz.pl (R.A.W.); bozena.rokita@p.lodz.pl (B.R.); 3Student Scientific Group, Department of Pharmaceutical Technology, Medical University of Białystok, Mickiewicza 2c, 15-222 Białystok, Poland; kaja.simonikk@gmail.com (W.K.S.); vkarolinaborkowskav@gmail.com (K.B.); 4Drug Transport and Delivery Research Group, Department of Pharmacy, Faculty of Health Sciences, UiT The Arctic University of Norway, Universitetsvegen 57, 9037 Tromsø, Norway; silje.mork@uit.no (S.M.); natasa.skalko-basnet@uit.no (N.Š.-B.)

**Keywords:** atorvastatin calcium, polyelectrolyte complex, freeze-drying, oral candidiasis, polysaccharide hydrogel, chitosan, tragacanth, xanthan gum, pectin, κ-carrageenan

## Abstract

Atorvastatin calcium, an antifungal agent, has the potential to be repositioned/repurposed to combat the increasing antimicrobial resistance. However, one of the most crucial issues in developing atorvastatin calcium-loaded products with a topical antifungal effect is achieving the optimal release and dissolution rates of this statin to produce the desired therapeutic effect. In this paper, we report on the development and pharmaceutical assessment of hydrogels composed of low-molecular-weight chitosan, tragacanth, and xanthan gum/pectin/κ-carrageenan as potential drug carriers for atorvastatin calcium for buccal delivery. Multidirectional analysis of the carriers with regard to their drug-release profiles and mucoadhesive, antimicrobial, and cytotoxic properties was accompanied by an evaluation of the freeze-drying process used to improve the hydrogels’ applicability. Using differential scanning calorimetry, Fourier transform infrared spectroscopy, and scanning electron microscopy techniques, the role of lyophilization in enhancing atorvastatin calcium delivery from polyelectrolyte complex-based matrices via drug amorphization was demonstrated. The freeze-dried hydrogels had significantly improved release and dissolution rates for the amorphic statin. Therefore, there is great potential for the use of lyophilization in the design of polyelectrolyte complex-based semi-solids in usable dosage forms for numerous crystalline and poorly water-soluble active substances.

## 1. Introduction

Atorvastatin (AT) is a synthetic, lipophilic statin—a cholesterol-lowering agent that inhibits 3-hydroxy-3-methyl-glutaryl-coenzyme A (HMG-CoA) reductase, which is responsible for endogenous cholesterol synthesis [1]. Marketed AT products are registered as the first-line therapy for dyslipidemia in patients at a moderate-to-high risk of developing cardiovascular diseases. Their therapeutic success arises from their pleiotropic activity, including anti-inflammatory, antioxidative, and antiproliferative effects on the vascular endothelium. Recent microbiological studies have proved that AT also has an antifungal effect, due to its ability to decrease the level of pathogenic ergosterol (a fungal cell membrane compound) [2,3,4,5,6,7]. In the literature, the fungicidal activity of AT has mainly been highlighted with respect to *Candida albicans*. Furthermore, the applicability of this statin in the local therapy of periodontitis or irinotecan-induced mucositis has also been confirmed [8,9].

Oral candidiasis (oral thrush) is an oromucosal fungal infection caused by pathogenic forms of *Candida*. The clinical manifestations of this condition can be different, which has resulted in a very comprehensive oral candidiasis classification system [10]. The individuals particularly at risk of colonization by pathogenic species of *Candida* are immunocompromised oncological and HIV-infected patients, people with coexisting dental or buccal mucosa disorders, those with insufficient oral hygiene or a hormonal imbalance, patients being treated with antibiotics/corticosteroids, and children. *C. albicans* is a ubiquitous commensal organism responsible for approximately 95% of oral thrush cases. This species is known for its high capacity to adapt to a human host and its variety of virulence factors, which enable a quick transition to its pathogenic form [10].

Hydrogels are dosage forms that are particularly useful in oromucosal drug application. Moreover, through their function as drug reservoirs, they can improve the efficiency of tissue-healing processes [11]. Polyelectrolyte complex (polycomplex, PEC)-based hydrogels—formulations composed of oppositely charged polyelectrolytes—are unique systems with great potential in drug administration, biomedical engineering, and chemical technology [12]. While there is a wide range of natural or synthetic polymers that can be used as the anionic component of a PEC, the assortment of available cationic hydrogel materials is much smaller [13]. Chitosan (CH), which has positively charged amine groups, is the most commonly utilized polycation in pharmaceutical technology. The cationic character of CH determines its ability to form PECs and the multidirectional biological activity for which the polymer is widely known. Besides hemostatic and anti-inflammatory activities, the amine groups of CH are responsible for its antimicrobial behavior. As mentioned above, combining CH with anionic macromolecules to form PECs is a common procedure to produce a unique drug carrier with new physicochemical and therapeutic characteristics. At the polycomplex design stage, the main goal is often the preservation of CH’s therapeutic effects alongside ameliorating the most problematic physicochemical properties of this polycation, such as its insufficient mechanical strength or the uncontrolled water uptake of CH materials.

The number of possible polycation–polyanion combinations seems to be unlimited. However, a significant issue in manufacturing PEC-based drug carriers is often their low homogeneity—a result of PEC particles’ precipitation and aggregation, which occur naturally after mixing with oppositely charged macromolecules. Among the various polycation–polyanion combinations, hydrogels composed of low-molecular-weight CH and tragacanth gum (TR) have been the objects of our interest, mainly due to their beneficial mechanical and mucoadhesive properties, which are important for buccally applied systems [14,15]. Herein, we explore combinations of CH and TR (as the dominant polyanion) with regard to their potential utility as carriers for the calcium salt of AT (ATC). The anionic component of the designed and formulated gels was enriched with three polysaccharides of natural origin: low-methoxy amidated pectin (LPE), kappa-carrageenan (κCA), and xanthan gum (XA). XA is an example of a prolonged-release polymer and viscosity enhancer, which has a beneficial impact on the mechanical and rheological properties of TR/CH hydrogels, as demonstrated in our previous work [14,15]. Furthermore, similar to the other two stabilizing agents LPE and κCA, XA is a polymer that is particularly useful in locally administered products due to its high mucoadhesiveness [16]. The ability of LPE to jellify in the presence of ubiquitous calcium ions makes it useful in providing a high drug loading and prolonged release or swelling control [17]. κCA is a representative of carrageenans, which have highly desirable physicochemical features and biological activities for pharmaceutical use, including anti-inflammatory, anticancer, and antiviral properties. Upon cooling or in the presence of potassium/calcium ions, it forms thermotropic and ionotropic hydrogels with tunable viscoelastic and mucoadhesive properties that are beneficial for controlled drug delivery [18]. The impact of the selected accompanying polyanions on the carrier’s performance was investigated and discussed, as well as the usability of the PEC gels as water-insoluble drug reservoirs. ATC represents a Biopharmaceutical Classification System (BCS) class II drug. It is a crystalline powder that is practically insoluble in aqueous solutions with a pH ≤ 4; very slightly soluble in distilled water, phosphate buffers with a pH of 7.4, and acetonitrile; slightly soluble in ethanol; and freely soluble in methanol and dimethyl sulfoxide (DMSO) [19,20]. As the solubility and rate of dissolution of crystalline ATC are low and often not sufficient for the desired therapeutic effect, many efforts have been made to improve these properties [20,21].

Freeze-drying (lyophilization or cryodesiccation) is a dehydration procedure that typically serves to preserve water-sensitive materials. It is routinely used to enable a detailed analysis of polymeric compositions via, for example, thermal or spectral measurements, which requires materials to be solid to be analyzed [14,15]. Here, lyophilization produced promising AT reservoirs, and those were subjected to the pharmaceutical evaluation alongside the initially designed and prepared PEC-based gels. This study focuses on the application properties of the formulated carriers, including their antimicrobial, mechanical, mucoadhesive, and cytotoxic properties, as well as their drug release profile. The report aims to highlight the strengths and weaknesses of the obtained AT gels and to indicate the potential usability of the lyophilization technique in eliminating the latter. Using AT aligns with drug repositioning/repurposing strategies, which aim to find new pharmaceutical applications of already approved active substances. Furthermore, in the era of increasing antibiotic resistance, it perfectly corresponds with the idea of exploring new antimicrobial strategies to eradicate recurrent fungal infections that are resistant to treatment [22]. AT has been confirmed to have synergistic antifungal effects with azole drugs (clotrimazole, ketoconazole, fluconazole, and itraconazole) against *C. albicans* [23]. Thus, through combining AT with the aforementioned commonly utilized antifungal agents, we could maximize the anti-*C. albicans* activity and reduce the risk of the emergence of resistant strains.

## 2. Results and Discussion

### 2.1. Characterization of Hydrogels and Lyophilizates

The pH of the prepared hydrogels (Table 1) was within the range of 5.64–6.25, which corresponds to the physiological pH of the oral cavity [24]. According to the calculated entrapment efficiency (%EE) values (Table 2), the ATC loading corresponded to the theoretical drug content within the hydrogels and lyophilizates. As for the tested mechanical parameters, the incorporation of ATC did not affect the firmness, cohesiveness, or mucoadhesiveness of the hydrogels (H1–H3 versus H1PL–H3PL) (Figure 1). While firmness describes the force required to compress the hydrogel, cohesiveness reflects the work of a probe that is needed for this deformation and adhesiveness corresponds to the force need to make the material adhere after application. The introduction of the active substance did improve the hydrogels’ retentivity (Section 2.7). Despite distinct differences in viscosity (Table 2), the formulations had comparable firmness and adhesiveness. Slightly lower values for the cohesiveness parameter were noted for the H2 gel. The viscosity values were not affected by drug incorporation.

The lyophilizates showed moisture contents within the range of 2.8 ± 2.1–8.1 ± 3.1% and %EE values as high as those of their hydrated counterparts.

### 2.2. In Vitro Release Assay

In vitro drug release is one of the most important issues in pharmaceutical evaluations of developed dosage forms, as it points to the applicability of the carrier in achieving the intended drug release profile. In vitro/in vivo correlations strongly depend on the established measurement procedure.

Determining the ATC release profile of the hydrogel matrices was not straightforward. In fact, the initial results of the HPLC analysis were unrepeatable and incoherent when samples of an acceptor medium were analyzed directly after sampling. Despite the provided sink conditions, which were based on the ATC solubility test results, the PEC-based carriers strongly hampered the release and dissolution of the active substance into the simulated saliva solution (SSS), likely due to strong drug–polymer and polymer–polymer interactions. It was assumed that ionic bonds between the cationic CH and the calcium salt of AT could significantly increase the inclusion of the drug into insoluble PEC particles. In fact, diluting the collected samples with a mixture of acetonitrile and a potassium dihydrogen phosphate buffer (pH 3.0) (60:40, *w/w*) as the mobile phase for the HPLC analysis (Section 3.2.2) enabled us to obtain measurable and repeatable values for the ATC concentration. The implemented modifications of the SSS, either by enriching it with PEG 400/Tween 80—solubilizer and surface active agents, respectively—or acidifying it, were fruitless since these modifications interfered with the HPLC data collection. With regard to acidifying the medium, our previous research pointed to a significant impact of pH change (to reflect the oral cavity conditions before and after a meal [24]) on CH dissolution and the subsequent disruption of CH PECs [25], but that impact was not observed here. As the tested solubility of ATC in acetonitrile was as low as for the aqueous solutions, disrupting the PEC particles using the mixture of acetonitrile and a potassium dihydrogen phosphate buffer (pH 3.0) was considered key for successful ATC liberation. Furthermore, in addition to the low pH of the phosphate buffer, the impact of its ionic strength on the stability of drug–polymer or polymer–polymer bonds must be noted [13]. The release profiles thus obtained showed the fastest ATC liberation and dissolution from the H1 and H3 gels. Similar release profiles were noted for the other gels, irrespective of the accompanying polyanion used, and approx. 80% of the ATC was collected after 4 h (Figure 2). The H2 gel with LPE was characterized by prolonged ATC release.

The obtained results from the in vitro release assay pointed to the necessity of distinguishing the drug release test from the drug dissolution test for ATC, as it is a practically water-insoluble substance. It is widely known that, for any drug to be absorbed, it must be present in a dissolved form. Otherwise, its bioavailability is inadequate and only high doses can provide the desired pharmacological effect. In the case of topically acting active substances, their low bioavailability is irrelevant and even preferable since the probability of systemic toxic events significantly decreases. Nonetheless, the essential issue is to provide a drug carrier that is highly miscible with physiological fluids at the site of local administration or has the ability to penetrate the biofilm when an antimicrobial effect is desired. In response to the difficulties in providing optimal ATC delivery from PEC hydrogels, the freeze-dried materials were also subjected to the in vitro release test. In comparison to the initially prepared hydrogels, the lyophilizates were characterized by significantly improved levels of ATC (Figure 3). Among all the sponge-like materials, LH1 had the most prolonged ATC release profile. ATC was detected in the acceptor medium for 30 min and only 29.6 ± 1.2% of the drug was detected after 6 h. On the other hand, LH3 provided the fastest drug liberation, achieving 68.5 ± 4.2% at the end point. It is assumed that the internal structure of the freeze-dried carriers had a strong impact on the release profiles. While the compact structure of LH1 might have affected the limited liberation of ATC enclosed in TG–CH and/or XG–CH complexes, the porous LH3 sponges facilitated the absorption and penetration of the dissolution medium, with a subsequent moderate swelling effect favoring a gradual leaching of the drug (Section 2.6).

### 2.3. Fourier-Transform Infrared (FTIR) Spectroscopy

FTIR analysis was performed to establish the role of ATC in the CH–gum polyelectrolyte complex formation process and the impact of the polymers on ATC performance. For pure ATC, bands that are characteristic of its crystalline form were identified (Figure 4). The transmission between 3600 cm^−1^ and 3700 cm^−1^ was attributed to the O–H stretching typical for trihydrated ATC [20,26]. The peaks at 3362 cm^−1^, 3265 cm^−1^, and 2921 cm^−1^ were attributed to the N–H, asymmetric O–H, and symmetric O–H stretching of the ATC molecules, respectively [20,26]. The absence or displacement of these peaks in the drug-loaded lyophilizates LH1–LH3 most likely resulted from ATC amorphization and/or its inclusion in the polymeric matrix; these observations were consistent with the in vitro release assay, differential scanning calorimetry (DSC), and scanning electron microscopy (SEM) results discussed below. The results did not indicate large differences between the spectra recorded for the placebo and drug-loaded lyophilizates, suggesting a high “miscibility” of the ATC in the polymers. In general, the absorption bands at around 3000–3500 cm^−1^ are typical for the hydroxyl groups present in the polysaccharide polymers (CH, TG, XG, LPE, and κCA) or residual water [27,28]. Any deviations in those bands could therefore result from either drug–polymer or polymer–polymer hydrogen bonding. The presence of the band at approx. 1600 cm^−1^, which is a characteristic of asymmetric bending vibrations of the amine groups in CH, might in turn indicate PEC formation due to the availability of NH^3+^. Their interaction with carboxyl (TG, XG, or LPE) or sulfate groups (κCA) could not be ignored due to the presence of bands characteristic of C=O, C–O, or S=O stretching (1760–1665 cm^−1^, 1320–1000 cm^−1^, or 1210–1260 cm^−1^ [29]); however, the overlapping bands of ATC did not allow for the identification of the shifts resulting from such interactions.

### 2.4. Thermal Characterization

DSC measurements were used to reveal the thermal behavior of the freeze-dried hydrogels. For the raw ATC, a strong endothermic peak characteristic of a melting process was indicated at 161 °C, confirming the crystalline nature of the drug [21,26,30] (Figure 5A–C). Subsequent cooling and a second round of heating did not lead to ATC recrystallization (Appendix A). Besides very wide endothermic peaks typical of moisture loss under gradual heating, the DSC curves of the placebo formulations LH1PL and LH2PL revealed an additional heat capacity drop, indicating a polymer glass transition (at 66.29 °C and 65.91 °C, respectively) [31]. There was an absence of a sharp endothermic event at the ATC melting temperature (T_m_) for the drug-loaded lyophilizates LH1–LH3. Instead, less or more pronounced wide endothermic peaks at 109.72 °C, 129.24 °C, and 106.85 °C for LH1, LH2, and LH3, respectively, were observed, which are characteristic of the glass transition temperature (T_g_) of amorphous ATC [21,32]. Furthermore, the second heating did not reveal any exothermic peaks specific to drug recrystallization, which might indicate the stability of the amorphous ATC form, just like the raw drug (Appendix A). The thermograms obtained for the physical mixtures of ATC and the polymers appeared to resemble the DSC curve for the pure drug; however, they were in a more repressed form, with strong endothermic events due to moisture evaporation (Figure 5D–F). Considering the substantial variations in ATC liberation from the fully hydrated and lyophilized formulations (Section 2.2), it could be assumed that the amorphous ATC was preserved in the solidified solvent during the freeze-drying [21,33]. PEC structures have been demonstrated to increase the solubility and/or bioavailability of different active substance enclosed in matrices composed of oppositely charged polyelectrolytes [34], and here, they were shown to also be important in preventing drug recrystallization and forming an amorphous solid dispersion (ASD).

### 2.5. Scanning Electron Microscopy (SEM) Analysis

SEM imaging showed some differences in the lyophilized hydrogels LH1–LH3 and their placebo counterparts (Figure 6). The LH1 and LH1PL materials were characterized by mostly uniform porous structures, while for the drug-loaded LH2 and LH3 and placebo LH2PL and LH3PL, heterogeneous architectures were observed. The addition of LPE (LH2 and LH2PL) and, to a lesser extent, κCA (LH3 and LH3PL) affected the presence of micro- and macropores, which disrupted the layered structure of the materials. Our previous investigations of PEC gels revealed a significant role of CH in triggering such a non-uniform structure [15]. According to the observations of Sungoradee and Srikulkit’s group [35], the interpolymer complexation between oppositely charged macromolecules (e.g., ATC and polymers) might result in the phase separation process affecting the formation of pores of different sizes. Nevertheless, neither crystalline nor amorphous drug forms were found to be freely dispersed in the polymeric matrices; this could be a consequence of the high miscibility of ATC with excipients compounded through ionic interactions due to ATC’s charge [32]. Small, roundish particles, which were more or less embedded in the lyophilizate walls, were most likely PEC particles, as they appeared in both placebo and drug-loaded materials. LH1 (TG/XG/CS), with its compact internal structure, had an exceptionally uniform and smooth surface compared to other tested materials.

### 2.6. Swelling Capacity

Upon contact with a solvent, solid polymers and polymer gels swell and dissolve at varying rates. Since swelling kinetics are regarded as an important parameter affecting the drug release profile or product disintegration, measurements of SSS uptake by both the lyophilizates and hydrogels were performed. Considering the noted variations in ATC release and solubility, verification of the solvent penetration rate depending on the polymer state (solid or gel) was crucial [36,37].

The fully hydrated materials showed negligible swelling abilities compared to the freeze-dried forms, and no significant variations were noted for the different samples (Figure 7A,B) [38]. The freeze-dried gels were characterized by intense and very dynamic swelling behaviors, which were particularly evident for the placebo materials (Figure 7B). In accordance with Fick’s law, the diffusion-controlled swelling affected the characteristic peak on the swelling curves, indicating rapid SSS uptake at the very beginning. Then, the relative stabilization of the sponge volumes was achieved, most likely due to the transformation of the polysaccharides from dry solids to soft gels with limited swelling [36].

Although LH1PL had the highest SSS uptake after the first 15 min, the addition of ATC resulted in a drop in the swelling capacity of about 50% (Figure 7B). This phenomenon was also observed for LH2, LH3, and their drug-free counterparts but the differences were not significant. It can be assumed that the differences in porosity characteristic between LH1 and LH1PL (SEM imaging, Section 2.5) were highly correlated with the observed swelling behaviors—the more porous material was more easily penetrated by the SSS. LH3 and LH3PL presented very similar swelling properties, while LH2PL had a more dynamic behavior when in contact with the SSS compared to LH2. The ATC carriers presented similar SSS absorption capacities after the first contact with the solvent. Then, a significant decline in SSS absorption capacity was registered for LH2. The heterogeneous structure of LH2 was likely responsible for the rapid and strong penetration of the SSS into the macropores scattered in the polymer matrix, whereas the gradual gelation accompanied by local viscosity growth led to the subsequent decrease in the swelling rate. Overall, the observed instant swelling of the lyophilizates in contact with the artificial saliva was found to be important for their mucoadhesiveness. Furthermore, the balance between SSS uptake and polymer matrix erosion observed in the swelling curves of the drug-loaded sponges indicates their potential resistance to the eroding activity of saliva in vivo.

### 2.7. Mucoadhesive Properties

One of the most important properties of buccal drug carriers is mucoadhesiveness. Retaining active substances at the site of action or absorption prevents unintended swallowing of the drug and provides a prolonged therapeutic effect. Another factor that limits the oromucosal residence time is saliva, a continuously produced eroding agent; therefore, it is necessary to develop a highly mucoadhesive carrier for buccal drug administration.

The strength of the interfacial joint with mucin gel and/or the residence time on porcine buccal mucosa were investigated. The lowest values of F_MAX_ (the force required to detach the upper probe from the mucin layer) and W_AD_ (the area under the force/distance curve) were observed for H1, whereas H2 and H3 revealed similar values for these parameters (Figure 8A,B). Moreover, the incorporation of ATC resulted in a substantial decline in H1’s mucoadhesiveness, which was not observed for the two other formulations. The exposure of the reference product to 10% mucin revealed values of F_MAX_ and W_AD_ that were several times higher than those of the developed gels. Interestingly, the mucoretention test did not confirm these observations, and the control had an almost negligible residence time—the samples started running down during application on the tissue (Figure 9A).

ATC-loaded H1–H3 revealed comparable mucoadhesion performances, while their placebo counterparts exhibited different behaviors. H1PL had outstanding mucoretention in comparison to H2PL and H3PL, reaching values equal to the residence time of H1–H3 (Figure 9B). The advantage of the H1PL sample over the other placebo gels was also highlighted by the texture analyzer measurements, which indicated a positive effect of XG on the gel’s mucoadhesiveness. XG is a commonly utilized pharmaceutical excipient for controlled and sustained drug release. In the presence of water, the dry polymer transforms into a hydrated state and subsequently, upon swelling, into a mucoadhesive, porous, and rubbery form that is sensitive to pH, ionic strength (especially Ca^2+^), and temperature [38]. As presented in Figure 9, a much shorter residence time was registered for the sponges. While LH2–LH3 and the placebo lyophilizates were characterized by negligible mucoretention, LH1 was washed off by the SSS after approx. 15 min. Since the measurement conditions were much more severe than those in the oral cavity, these results highlight the strong mucoadhesion behavior of the ATC gels and LH1.

### 2.8. Antimicrobial Activity

Overall, the ATC gels showed the strongest antifungal activity against *C. albicans*—the main causative agent of oral thrush (Figure 10A,B). *C. krusei* and *C. parapsilosis* were much more resistant to the tested formulations, and H2 did not affect the colony growth of *C. krusei* at all. The gels revealed stronger anti-*C. albicans* behaviors in comparison to a commercially available cream with clotrimazole. No antifungal effect was noted for the placebo hydrogels, which emphasizes the crucial role of ATC in reducing the growth of *Candida* strains. The strong ionic interactions between CH and different types of anionic polymers may have limited the ability of the polycation to bond with the negatively charged surface of fungal cells and exert an antimicrobial effect. Despite the established and widely discussed differences in ATC delivery from fully hydrated and freeze-dried carriers, the antimicrobial performance in this study, especially with regard to the *C. albicans* strain, was comparable for both dosage forms. The LH1 lyophilizate revealed slightly wider inhibition zones of *C. albicans* growth than the H1 gel. The sponges were also more effective at reducing *C. krusei* growth; however, they did not show any anti-*C. parapsilosis* activity, in contrast to the ATC gels. These results showed that despite the encountered problems in determining the manner of drug release from hydrogel carriers, the form and degree of binding with polymeric components of ATC affected its antifungal performance to a small degree. After 24 h of contact with *Candida* cultures, similar inhibition zones were observed for both dosage forms. It could be concluded that sufficiently long contact between ATC and the yeast is crucial for inhibiting its growth regardless of the ATC release rate.

Since *C. albicans* is an opportunistic pathogen that invades when there are changes in the host microenvironment that favor its proliferation, we also investigated the potential impact of the developed drug carriers on the physiological flora of the oral cavity. For this purpose, cultures of *Lactobacillus brevis* (a Gram (+) representative of the natural microbiota with a highly beneficial impact on oromucosal inflammation disorders) or *Streptococcus mutans* (a major pathogen of dental caries) were exposed to the hydrogels/sponges [39]. The observations from the pure ATC and ATC dissolved in DMSO (dimethyl sulfoxide) suggested that the incorporation of ATC into a PEC-based matrix protected *L. brevis* against the potentially harmful effects of ATC.

### 2.9. Cytotoxicity

The HUVEC (Human Primary Umbilical Vein Endothelial Cell) model used in investigating endothelial cell functions, such as inflammation, oxidative stress, hypoxia, and response to infection, was utilized to assess the cytotoxicity of the lyophilizates. HUVECs are relatively sensitive, e.g., much less resistant to toxins than fibroblasts, but they are suitable for investigating buccal products due to the importance of vasculo- and angiogenesis in oromucosal tissue differentiation [15,40].

The buccal mucosa is considered one of the most challenging routes for drug administration. Apart from oral cavity movements or exposure to food and liquids, the important factor limiting the residence time of buccally administered dosage forms is salivary secretion. While the unstimulated salivary flow rate ranges between 0.25 and 0.5 mL min^−1^, the stimulated flow rate is from 1 to even 5 mL min^−1^ during eating. An adult person produces approx. 500–2000 mL of saliva per day. On average, 1 mL of the fluid is present in the mouth before swallowing [41,42]. Thus, the protective and purifying properties of the saliva and mucus present on buccal epithelial cells should be taken into consideration when studying the cytotoxic effects of buccally applied products. For this reason, apart from the standard examination procedure, a modified MTT test was applied to more accurately reflect the actual oral cavity conditions: a shorter exposure of the cells to the prepared eluates and a greater range of ATC concentrations were used (Section 3.2.15). Compared to the standard protocol, which revealed a strong cytotoxic effect of ATC on the HUVECs after 48 h, a reduction in the sample–cell contact time significantly improved their viability, except for the undiluted LH1–LH3, which exhibited cytotoxic effects (Figure 11A,B). Diluting the eluates to match the in vivo conditions led to gradual leaching of the product and improved cell growth: a 25% dilution resulted in, on average, 80% cell viability, meeting the minimum 70% viability recommended by the FDA [43]. Besides the exposure time to different concentrations of ATC, the viscosity of the eluates (determined by either the polymer composition or degree of dilution) was equally important for their cytotoxic effect. In fact, LH2 prepared from the LPE-based hydrogel with the highest viscosity caused the greatest cell mortality among all the two-fold-diluted eluates, most likely due to the impeded transport of oxygen and nutrients necessary for metabolic processes and cell proliferation [44]. The results of the MTT test for ATC dissolved in DMSO indicated at least 70% cell viability at a drug concentration of 0.96 µM, which is equal to the ATC content in the eluates when diluted 32 times (3.13%) (Figure 11C). The 0.24 µM drug solution resulted in 100% viability, which is consistent with the literature showing that this ATC concentration does not affect HUVEC growth [45]. Significantly lower cell mortality was observed for the developed dosage forms in both procedures, highlighting a cytoprotective effect of the polymers and/or PEC structures—components of the carriers—on this cell line.

## 3. Materials and Methods

### 3.1. Materials

Low-molecular-weight chitosan (CH) (derived from snow crabs (*Chionoecetes opilio*), shrimps, or squids; degree of deacetylation: 79.9%; viscosity: 31–70 mPa·s at 20 °C for 1% CH in 1% acetic acid; average molecular weight: 232 kDa) was purchased from Heppe Medical CS GmbH (Haale, Germany). Tragacanth gum (TR) (from *Astragalus gummifer*; average molecular weight: 840 kDa), xanthan gum (XA) (from *Xanthomonas campestris*; average molecular weight: 1000 kDa), dimethyl sulfoxide (DMSO), and mucin type II from porcine stomach were obtained from Sigma-Aldrich (St. Louis, MO, USA). Low-methoxy amidated pectin (LPE) (type CF 010; degree of esterification: 34%; degree of amidation: 17%; pH: 4.2 for 2.5% solution in distilled water at 20 °C) was kindly donated by Herbstreith & Fox and GmbH & Co. KG (Neuenbürg, Germany). Atorvastatin in its calcium salt form (ATC) and κ-carrageenan (κCA) were purchased from Xi’an Kerui Biotechnology Co., Ltd. (Xi’an, China). Glycerol was obtained from Fagron (Kraków, Poland). Sodium benzoate, disodium hydrogen phosphate, potassium dihydrogen phosphate, and 85% lactic acid were obtained from Chempur (Piekary Śląskie, Poland). Acetonitrile and absolute ethanol were obtained from J. T. Baker (Phillipsburg, NJ, USA). Stock cultures of *C. albicans* (ATCC^®^ 10231)*, C. krusei* (ATCC^®^ 6528), *C. parapsilosis* (ATCC^®^ 22019), and *L. brevis* (ATCC^®^ 8287), as well as Sabouraud’s agar were purchased from Biomaxima (Lublin, Poland). MRS Agar was obtained from Pol-Aura (Morąg, Poland). Sterile 0.9% sodium chloride was obtained from Polpharma S. A. (Starogard Gdański, Poland). Nylon membrane filters (0.45 µm) were provided by Millipore (Billerica, MA, USA) and polyamid filters (0.45 µm) were provided by Sartorius Stedim Biotech GmbH (Goettingen, Germany). Elugel^®^ (chlorhexidine digluconate 0.2%; Pierre Fabre Medicament, Lavaur, France; series: 3GC7E; expiry date: 09.2026) and Clotrimazolum GSK^®^ cream (clotrimazole 10 mg/g; GSK PSC Poland, Poznań, Poland; series: 93JV: expiry date: 09.2026) were used as positive controls in, respectively, the mucoadhesion and antimicrobial tests. HUVECs (Human Primary Umbilical Vein Endothelial Cells; PCS-100-010) and Vascular Cell Basal Medium (PCS-100-030) supplemented with Endothelial Cell Growth Kit-VEGF (PCS-100-041) were obtained from the American Type Culture Collection (ATCC^®^, Manassas, VA, USA). 3-(4,5-dimethylthiazol-2-yl)-2,5-diphenyltetrazolium bromide (MTT), Triton X-100, phosphate-buffered saline (pH 7.4), and a stabilized penicillin/streptomycin solution (PS) were purchased from Sigma-Aldrich (Darmstadt, Germany).

Porcine buccal mucosa was obtained from the veterinary service of the local slaughterhouse (Turośń Kościelna, Poland). The tissue was dissected directly after killing an animal, subsequently rinsed with a sterile 0.9% sodium chloride solution, and stored at −20 °C until it was used. The acquisition of the tissues did not require the approval of the Bioethics Committee on Animal Testing.

The simulated saliva solution (SSS) composed of 0.1 M disodium hydrogen phosphate and 0.1 M potassium dihydrogen phosphate, following the protocol of Marques et al. [24] with modifications, was used for the research analyses.

### 3.2. Methods

#### 3.2.1. Preparation of Hydrogels and Lyophilizates

A 2.5% (*w/w*) CH solution was prepared by dispersing the polymer in 1% (*v/v*) lactic acid at 40 °C. The anionic components of the systems were obtained by gradually adding the polysaccharides to the Unguator^®^ tube after the initial moisturizing of the TR in a beaker filled with water and glycerol at a concentration of 5% (*w/w*), and then mixing it with Unguator Eprus^®^ U500 (Eprus Sp. z o.o., Bielsko-Biała, Poland) for 4 min at 1375 rpm. In the next step, the mixture was preserved by adding sodium benzoate dissolved in a small amount of water (0.1%, *w/w*). After 2 min of mixing at 1375 rpm, ATC solubilized in a two-fold-higher amount of polyethylene glycol 400 (PEG 400) was incorporated into the polyanion dispersion using the same mixing procedure. Then, a CH solution was gradually added to the tube (time: 2 min; speed: 1375 rpm). Placebo hydrogels were prepared as controls and named H1PL–H3PL (Table 1).

After 24 h of freezing at −75 °C, samples of the hydrogels were subjected to a lyophilization process (Freezine 6, Labconco, Kansas City, MO, USA). They were labeled as LH1–LH3 and LH1PL–LH3PL for the drug-loaded and placebo materials, respectively. The same freezing protocol was applied to all the samples.

#### 3.2.2. Quantification of ATC Using High-Performance Liquid Chromatography (HPLC)

A quantitative analysis of AT was carried out with an Agilent Technologies 1200 HPLC system equipped with a G1312A binary pump, G1316A thermostat, G1379B degasser, and G1315B diode array detector (Agilent Technologies, Waldbronn, Germany) using a Waters Spherisorb^®^ ODS, 4.6 mm × 250 mm, 5 µm column (Waters Corporation, Milford, MA, USA) [46]. The retention time was 6.47 min, and the obtained results were analyzed with Chemstation 6.0 software. A quantitative analysis was performed with the calibration curve y = 46.455x + 32.211 (R^2^ = 0.9987) over the concentration range of 5–100 µg mL^−1^.

#### 3.2.3. ATC Solubility Test

The solubility of ATC in the twenty-fold-diluted SSS was investigated according to the shake-flask method previously described for gels with secnidazole [15].

#### 3.2.4. ATC Entrapment Efficiency (%EE)

One gram of the H1/H2/H3 gels was placed in a plastic tube with 5 mL of ethanol and manually shaken for 5 min [15]. After centrifuging at 3000 rpm for 15 min, the obtained extract was decanted and another 5 mL of ethanol was added to the tube to repeat the extraction procedure. The collected supernatant was filtered with 0.45 µm nylon filters and diluted with the mobile phase. The ATC content in the lyophilizates was measured by placing approximately 20 mg of the material in a 25 mL flask with 10 mL of ethanol. After 24 h of shaking at 150 rpm in a water bath (37.0 ± 0.5 °C), the obtained extract was filtered with 0.45 µm nylon filters and diluted with the mobile phase for the ATC measurement. The %EE values for both dosage forms were calculated using the following formula:

%EE = Q_a_/Q_t_ × 100
(1) where Q_a_ is the actual drug content in the examined 1 g of the hydrogel/lyophilizate, and Q_t_ is theoretical drug content in a 1 g sample.

#### 3.2.5. pH Analysis

pH measurements were performed in triplicate at 25 ± 2 °C by immersing the glass electrode of an Orion 3 Star pH meter (Thermo Scientific, Waltham, MA, USA) at three different points in the analyzed hydrogel.

#### 3.2.6. Viscosity Measurements

The viscosity of the gels was measured at 25 ± 2 °C using a Brookfield viscometer (DVNXRVDJG, Brookfield Engineering Laboratories, Middleboro, MA, USA) equipped with a CPA52Z cone (Ø 24 mm, ∡ 3 °) at a shear rate of 2 s^−1^.

#### 3.2.7. Analysis of the Mechanical Properties: Firmness, Consistency, and Adhesiveness

Mechanical properties of the hydrogels were evaluated at 25 ± 2 °C using a TA.XT Plus Texture Analyzer (Stable MicroSystem, Godalming, UK) equipped with a 5 kg load cell and A/Be backward extrusion measuring system. The analysis was performed by pushing a plexiglass disc (Ø 25 mm) into a 30 g sample at a speed of 4 mm s^−1^, starting from a position right above the hydrogel surface. The distance and trigger force were 10 mm and 10 g, respectively. By using Texture Exponent 32 software, the mechanical parameters of firmness, consistency, and adhesiveness were calculated from the force versus time plot.

#### 3.2.8. In Vitro Drug Release Assay

The ATC release kinetics for approx. 0.75 g of each of the hydrogel samples were determined using an Erweka^®^ Dissolution Tester type DT 600HH (Erweka GmbH, Heusenstamm, Germany) and Teflon enhancer cells with a diffusion area of 3.8 cm^2^ at 75 rpm. To obtain the sink conditions, 600 mL of twenty-fold-diluted SSS was used as the dissolution medium. By diluting the SSS, the risk of interactions between the phosphate ions in the saliva and calcium ions of ATC, which could interfere with the HPLC data collection, was minimized. After 15, 30, 60, 120, and 240 min, a 1 mL sample of the SSS was withdrawn, filtered through nylon filters (0.45 µm), and analyzed via HPLC.

Samples (50 mg) of the prepared lyophilizates (sponges) were also subjected to an in vitro drug release test using an Erweka^®^ Dissolution Tester type DT 6 equipped with baskets, and 100 mL of the diluted SSS mixed at a speed of 75 rpm was enough to provide sink conditions. The quantitative analysis of ATC was performed as described above.

#### 3.2.9. FTIR

The freeze-dried carriers were subjected to FTIR analysis using a Thermo Nicolet Avatar 330 (Thermo Electron Corporation, Waltham, MA, USA). Potassium bromide was mixed with the sample in a mass ratio of 100:1 in an agate mortar in order to obtain a homogeneous powder, which was then pressed into a tablet form. The spectra of the samples were obtained in the transmission mode in the wavelength range of 4000–500 cm^−1^ with a resolution of 4 cm^−1^. The number of scans was 64.

#### 3.2.10. Thermal Analysis

Evaluation of thermal properties of the lyophilizates was performed using a DSC Q200 differential scanning calorimeter (TA Instruments, New Castle, DE, USA). The instrument was calibrated for both temperature and enthalpy using indium with a melting temperature and heat of fusion of 156.6 °C and 28.57 J g^−1^, respectively. Samples (approx. 3 mg) were sealed in aluminum pans and analyzed in the temperature range of 20–200 °C with a heating/cooling rate of 10 °C min^−1^, under a nitrogen flow of 50 mL min^−1^.

#### 3.2.11. SEM Imaging

The surface and internal structures of the freeze-dried gels were investigated using an SEM microscope (Hitachi TM-1000, High-Technologies Corporation, Tokyo, Japan). For this purpose, a representative fragment of the lyophilizate was placed on the SEM stand, coated with gold, and then analyzed.

#### 3.2.12. Swelling Behavior

Swelling tests were carried out with 15 mL vertical Franz diffusion cells (PermeGear V6-CB Diffusion System, Hellertown, PA, USA) by measuring the volume of the SSS taken up by a sample placed in the donor compartment at 37 ± 1 °C [47]. Approximately 20 mg or 200 mg of a lyophilizate or hydrogel, respectively, was weighed on a polyamide membrane (0.45 µm pore size) and then put in contact with the SSS. The amount of the medium absorbed by one milligram of a sample was measured by refilling the acceptor compartment with the missing SSS (via a 500 µL syringe; Hamilton, Switzerland) after 5, 10, 15, 30, 60, 120, and 240 min.

#### 3.2.13. Evaluation of Mucoadhesive Properties

The mucoadhesive properties of the hydrogels were assessed using two independent methods and two mucoadhesive models.

The ex vivo retention time on porcine buccal mucosa was determined according to the “wash off” method using a USP disintegration tester [48,49]. Tissue pieces (2 × 5 cm) were glued to the internal beaker wall above the line indicating a volume of 700 mL. The container was subsequently filled with 700 mL of the SSS and thermostatted at 37 ± 1 °C. A 0.5 g sample of a hydrogel was applied on the glued tissue and a plexiglass cylinder (Ø 6 cm, 280 g) was installed. Its vertical movements enabled repetitive contact between the SSS and the sample; the point at which the entire hydrogel was removed from the tissue was the end point of the experiment. The analysis was carried out using a commercially available dental gel with chlorhexidine digluconate as the positive control and cellulose paper as the negative control. Using the same procedure, cylindrical samples of the lyophilizates (Ø 8 mm) were also evaluated.

In the mucoadhesion assay using a TA.XT Plus Texture Analyzer (Stable Microsystems, Godalming, UK), approx. 0.5 g of a hydrogel sample was placed on the upper probe of the instrument. A 2 mL volume of a 10% (*w/w*) mucin dispersion was inserted into the A/Muc mucoadhesion ring and thermostatted at 37 ± 2 °C. At 1 min after weighing the sample, the upper probe was installed on the travelling arm and lowered at a constant speed of 0.1 mm/s. The sample was kept in contact with mucin for 120 s under a force of 0.5 N, and then the interacting surfaces were separated at speed of 0.1 mm/s. The parameters F_MAX_ and W_AD_ were calculated using Texture Exponent 32 software.

W_AD_ = A × 0.1 × 1000
(2)
where A is the area under the force/distance curve. The multiplication by 0.1 converts the time measurement to distance, while multiplication by 1000 enables the parameter to be expressed in µJ [50].

#### 3.2.14. Antimicrobial Activity Test

The anti-*Candida* activity of the hydrogels and sponges were assessed using three selected strains, *C. albicans* (ATCC^®^ 10231), *C. krusei* (ATCC^®^ 6528), and *C. parapsilosis* (ATCC^®^ 22019), and the plate diffusion method of the Clinical and Laboratory Standards Institute (CLSI) [51]. The fungal/bacterial inoculum was prepared using sterile 0.9% sodium chloride to achieve a suspension with a density of 0.5 on the McFarland (MF) scale (approx. 1.5 × 10^6^ colony-forming units per milliliter). Each sample was placed in a 7 mm well in Sabouraud’s agar, which had been inoculated with 100 μL of a specific broth culture beforehand. The plates were incubated at 37.0 ± 0.1 °C for 24 h (*C. albicans* and *C. krusei*) or 48 h (*C. parapsilosis*) and, after this time, the diameters of the inhibition zones were measured [52]. The impact of the commercially available cream with 1% clotrimazole, 1.5% (*w/w*) solution of ATC in DMSO (with pre-determined no effect on the fungi growth), and ATC powder on *Candida* growth was also assessed.

The protective effect of the developed formulations on *L. brevis* (ATCC^®^ 8287) was examined according to the above method, but using MRS Agar.

#### 3.2.15. Cytotoxicity Testing

Cytotoxicity studies were performed on the HUVEC line in Vascular Cell Basal Medium supplemented with Endothelial Cell Growth Kit-VEGF and 1% antibiotic penicillin/streptomycin. The tests were performed on cells between passages 3 and 7. During these studies, the cells did not show any signs of microbiological contamination, and no change in cell morphology was observed. On each day, at the beginning of the analytical procedure, the culture was at 90% confluence.

##### Preparation of Eluates for Cytotoxicity Studies

The eluates of the test material were prepared based on the guidelines of ISO 10993:12 [53]. For this purpose, the test material was weighed (under sterile conditions) and the appropriate volume of medium was added, maintaining a ratio of 0.1 g of test material/1 mL of culture medium (this was the initial eluate of the tested material, i.e., a concentration of 100%). Then, the eluate was subjected to 24 h of incubation at 37 °C and shaking at 200 rpm. Following that, a series of appropriate dilutions of the initial eluate was prepared using the medium.

##### MTT Assay

Cells were seeded in 96-well plates at a density of 6 × 10^3^ cells per well and cultured in 5% CO_2_ at 37.0 ± 0.5 °C. The next day, the test material eluates prepared in a series of appropriate dilutions from 100% to 0.39% were added to the culture. Cells not exposed to any stimulating factor were treated as the negative control (NC), while cultures administered Triton X-100 at a final concentration of 0.01% 24 h before the addition of the MTT reagent were used as the positive control (PC). The cells that were exposed to the test material for 48 h (the standard examination procedure) or 1 h were rinsed with medium, supplemented with fresh medium, and incubated for 47 h. After the incubation period, the MTT reagent was added to the culture at a concentration of 3 mg/mL (50 μL/well) and incubated for another 4 h. After this time, the supernatant was removed from the culture, and 100 μL of 70% isopropanol/HCl was added to dissolve the precipitated crystals. Absorbance was measured using a microplate reader (BioTek, Boston Industries, Walpole, MA, USA) at 570 nm and 650 nm. The results are presented as % cell viability relative to untreated control cells (NC), for which we give the mean value ± standard error of the mean for at least three independent biological replicates, with eight runs in each biological replicate.

#### 3.2.16. Statistical Analysis

The statistical data analysis was performed using a one-way ANOVA with *p* < 0.05 as the threshold for significance. All calculations were performed using Microsoft Excel^®^.

## 4. Conclusions

One of the most important issues in developing ATC-loaded products as topical antifungal drugs is to achieve the optimal drug release and dissolution necessary for their pharmacological effect. The inclusion of ATC in PEC-based carriers raises an additional challenge with the risk of drug–polymer or polymer–polymer ionic interactions, which limit liberation of the drug. Herein, freeze-drying of hydrogels composed of cationic CH and anionic polysaccharides was shown to be an effective method for enhancing ATC release due to stabilization of the amorphous drug in the polymer matrices. The impact of the lyophilization parameters on the sustainability of polycation–polyanion or drug–polymers interactions cannot be overlooked in the context of the enhanced solubility of ATC. Despite their high mucoadhesive properties and desired texture, the formulated hydrogels presented incoherent results in the in vitro drug release test. Their freeze-dried counterparts demonstrated significantly improved delivery of ATC, which prompted more of a focus on the lyophilized form of the semi-solids as promising carriers for ATC—a representative of crystalline drugs with low water solubility. This study assessed the most important application properties of the lyophilizates. The priority of further studies should be improving the ATC delivery rate from the designed PEC carriers to the buccal mucosa, as well as other routes of application (vaginal mucosa, in particular, considering the common and recurrent problem of vaginal *Candida* infections).

## Figures and Tables

**Figure 1 ijms-26-02267-f001:**
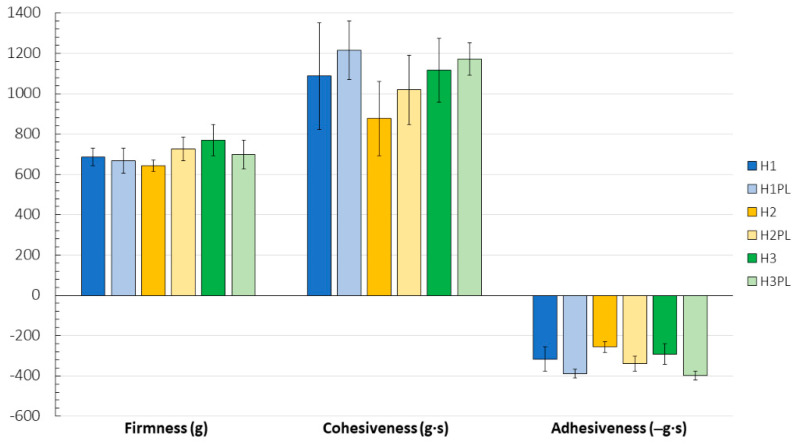
Mechanical properties of hydrogels H1–H3 and their placebo counterparts H1PL–H3PL (mean ± SD, n ≥ 5).

**Figure 2 ijms-26-02267-f002:**
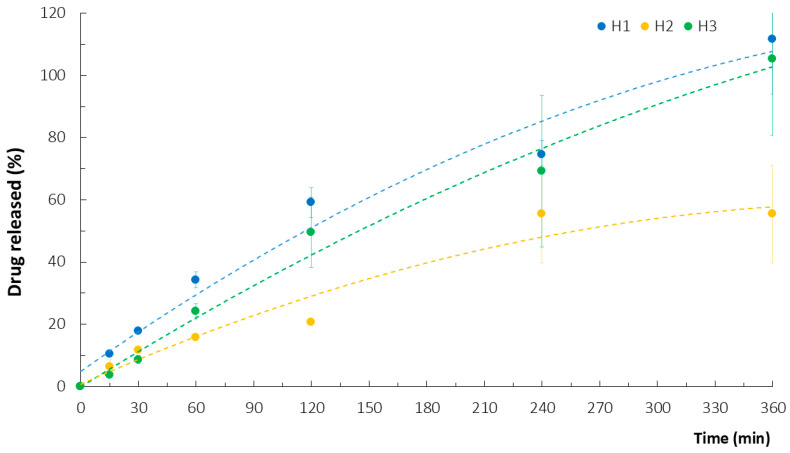
In vitro ATC release from hydrogels H1–H3 with corresponding polynomial trend lines (mean ± SD, n = 3, *p* < 0.05).

**Figure 3 ijms-26-02267-f003:**
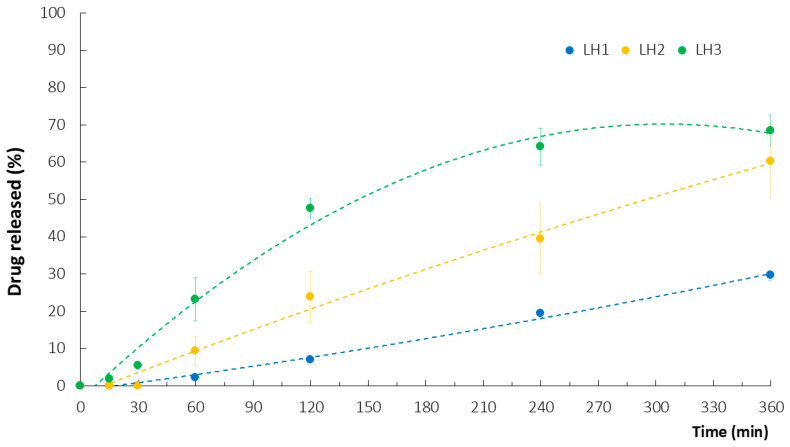
In vitro ATC release from lyophilizates LH1–LH3 with corresponding polynomial trend lines (mean ± SD, n = 3, *p* < 0.05).

**Figure 4 ijms-26-02267-f004:**
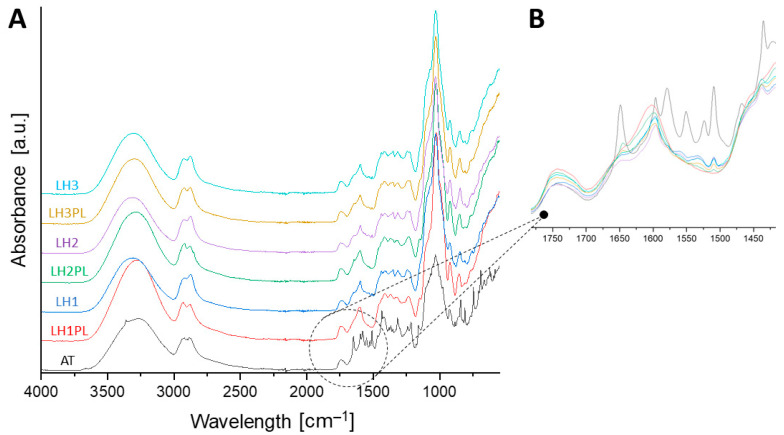
FTIR spectra of the drug-loaded and placebo lyophilizates in the wavelength range of 4000–500 cm^−1^ (**A**), with a magnified view of the section between 1775 cm^−1^ and 1425 cm^−1^ (**B**).

**Figure 5 ijms-26-02267-f005:**
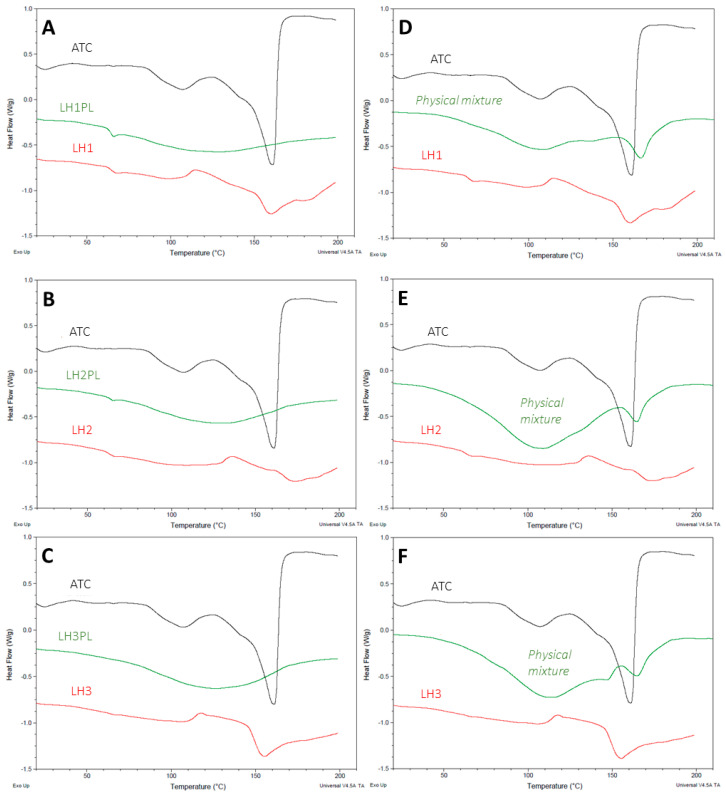
DSC thermograms (exothermic) obtained at first heating for ATC-loaded and placebo samples (**A**–**C**). Thermal behaviors of atorvastatin calcium (ATC) and LH1–LH3 and their respective physical mixtures (**D**–**F**). Cooling and second heating effects are presented in the figures included in the Appendix A.

**Figure 6 ijms-26-02267-f006:**
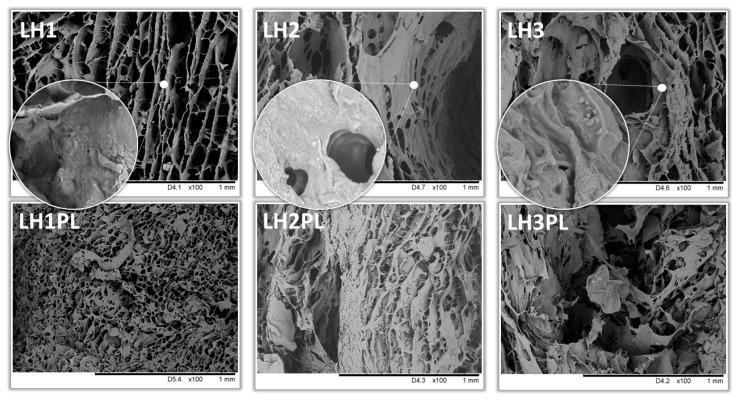
Selected SEM images of the freeze-dried hydrogels (LH1–LH3 and LH1PL–LH3PL) under 100× (main pictures) and 1000× (magnified sections) magnification.

**Figure 7 ijms-26-02267-f007:**
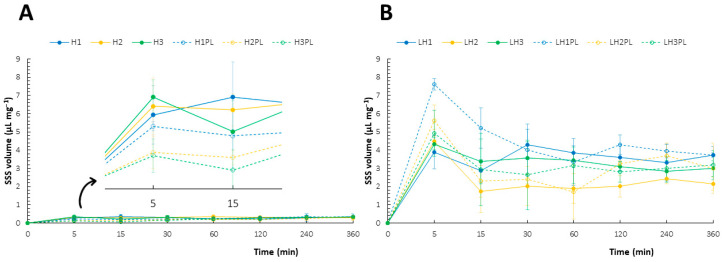
Swelling behaviors of the hydrogels (**A**) and lyophilizates (**B**) in the simulated saliva solution (mean ± SD, n = 3).

**Figure 8 ijms-26-02267-f008:**
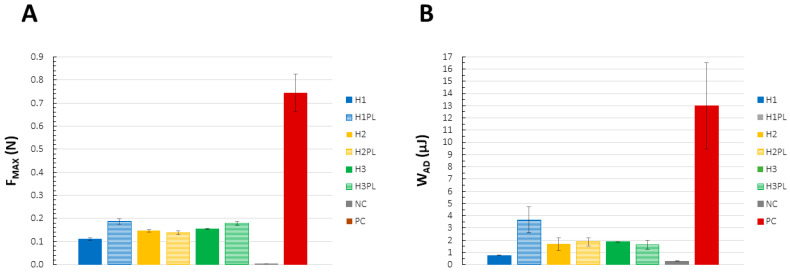
Mucoadhesive properties of the hydrogels (H1–H3, H1PL–H3PL), expressed as the (**A**) mucoadhesion force (F_MAX_, N) and (**B**) work of adhesion (W_AD_, µJ) (NC (negative control): cellulose paper; PC (positive control): Elugel^®^); mean ± SD, n ≥ 3.

**Figure 9 ijms-26-02267-f009:**
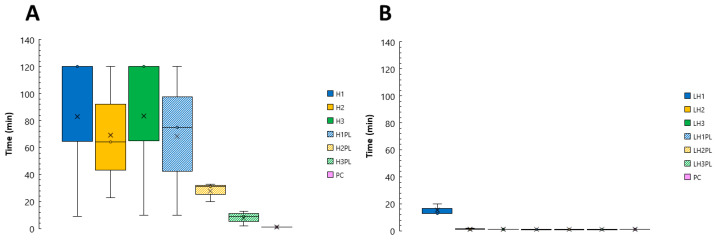
Mucoretention of the hydrogels (**A**) and lyophilizates (**B**) (PC (positive control): Elugel^®^); n = 3.

**Figure 10 ijms-26-02267-f010:**
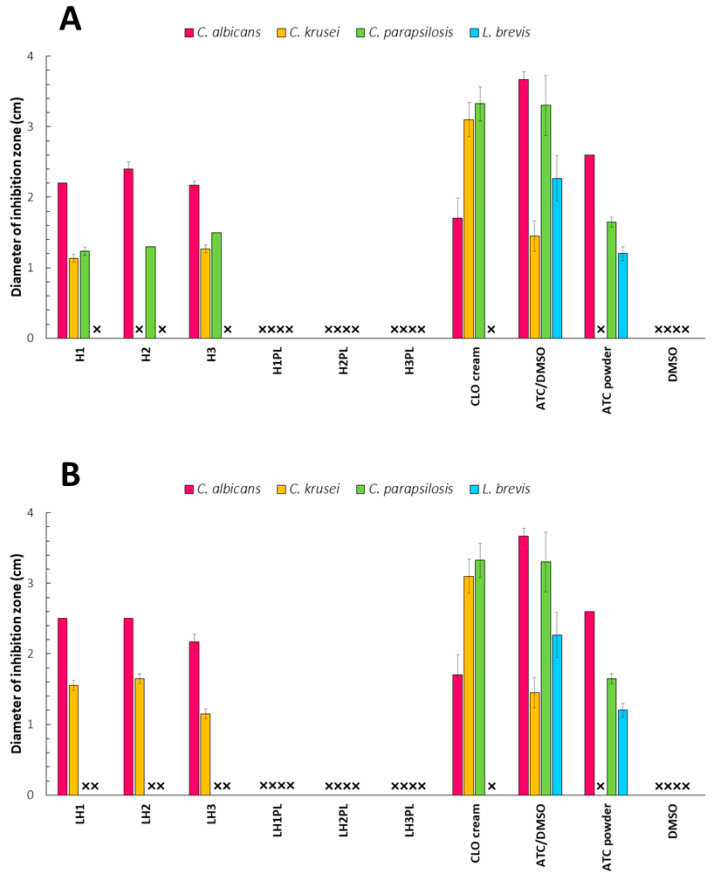
Antimicrobial activities of the hydrogels (**A**) and lyophilizates (**B**) against *Candida albicans*, *C. krusei*, *C. parapsilosis*, and *Lactobacillus brevis* (controls: CLO cream with 1% (*w/w*) clotrimazole, ATC/DMSO, and ATC powder (atorvastatin calcium dissolved in DMSO or as a pure crystalline substance); mean ± SD, n = 3. The symbol “×” indicates no antimicrobial effect.

**Figure 11 ijms-26-02267-f011:**
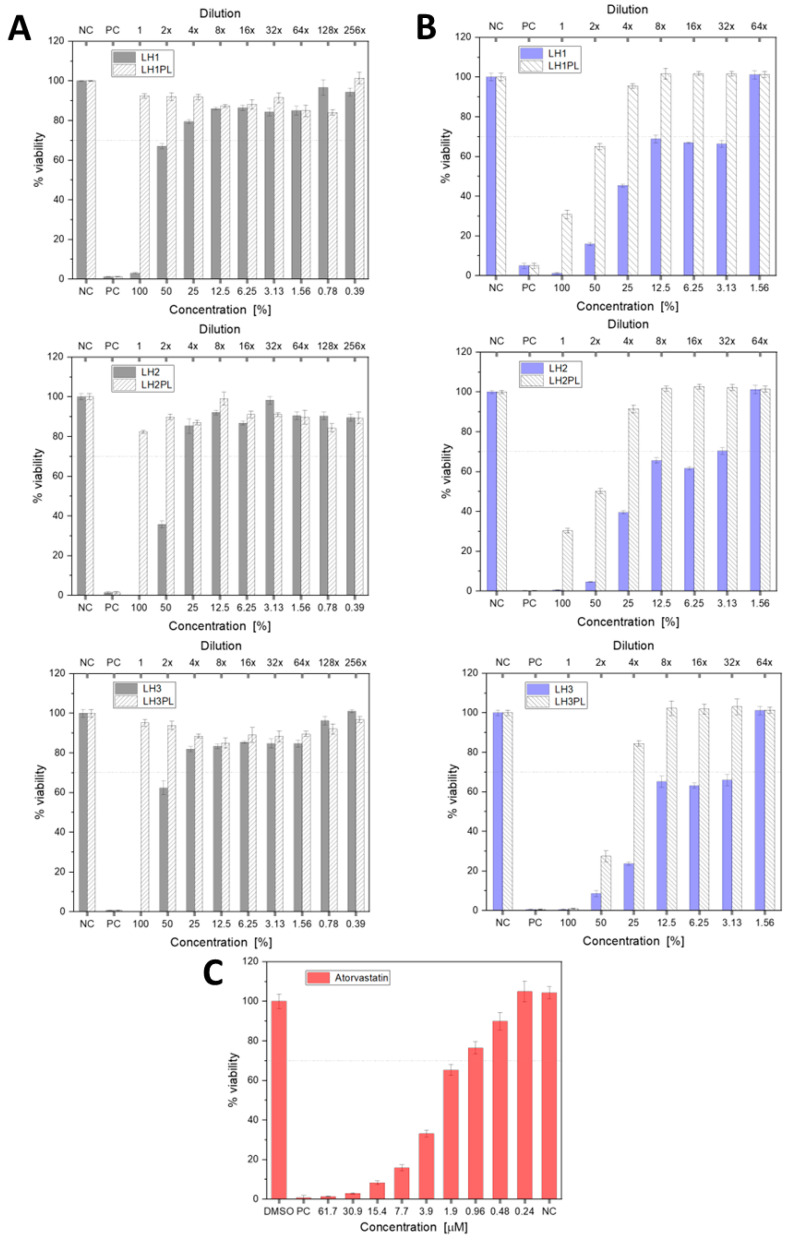
Cell viability according to the modified (**A**) and standard MTT procedures (**B**) with respect to the positive (PC) and negative controls (NC: untreated cells). Cytotoxicity of the pure drug dissolved in DMSO (**C**). Mean cell viability ± SD, n = 8.

**Table 1 ijms-26-02267-t001:** Composition of the prepared hydrogels.

Hydrogel	Components (%)								
	Polymeric Carrier								ATC
	TR	XA	LPE	κCA	CH	Glycerol	PEG 400	SB	
H1	4.0	0.4	×	×	0.4	5.0	3.0	0.04	1.5
H2	4.0	×	0.4	×	0.4	5.0	3.0	0.04	1.5
H3	4.0	×	×	0.4	0.4	5.0	3.0	0.04	1.5
H1PL	4.0	0.4	×	×	0.4	5.0	3.0	0.04	×
H2PL	4.0	×	0.4	×	0.4	5.0	3.0	0.04	×
H3PL	4.0	×	×	0.4	0.4	5.0	3.0	0.04	×

TR—tragacanth gum; XA—xanthan gum; LPE—low-methoxy amidated pectin; κCA—kappa-carrageenan; CH—chitosan; PEG 400—polyethylene glycol 400; SB—sodium benzoate; ATC—atorvastatin calcium.

**Table 2 ijms-26-02267-t002:** Characterization of the developed ATC carriers: hydrogels (pH, entrapment efficiency (%EE), and viscosity) and lyophilizates (%EE) (mean ± SD, n = 3).

Hydrogel	pH	%EE	Viscosity (cP)
H1	5.95 ± 0.03	118.2 ± 3.3	75 707 ± 3 698
H2	6.21 ± 0.04	110.7 ± 3.1	116 800 ± 2 343
H3	5.66 ± 0.02	100.3 ± 14.9	87 213 ± 3 379
LH1	×	107.2 ± 7.0	×
LH2	×	117.8 ± 7.3	×
LH3	×	100.8 ± 12.2	×

## Data Availability

The data that support the findings of this study are available from the corresponding author upon reasonable request.

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
