# Peer review of "Improving Atorvastatin Release from Polyelectrolyte Complex-Based Hydrogels Using Freeze-Drying: Formulation and Pharmaceutical Assessment of a Novel Delivery System for Oral Candidiasis Treatment"

_ijms, 2025, doi:10.3390/ijms26052267_

Round 1
Reviewer 1 Report
Comments and Suggestions for Authors
The paper by PotaÅ›-Stobiecka et al. describe the use of (lyophilised) hydrogels for the delivery of a model drug, ATC, for the treatment of oral candidiasis. The study is interesting and well-structured, reporting the characterisation of the gels and the evaluation of their release ability, antibacterial activity and cytotoxicity. The study is of potential interest for the readers of IJMS, provided that the following comments are addressed.
- Declare acronyms when first used in the text, e.g., %EE
- To improve readability of the results, shortly describe the chemistry of the hydrogels at the beginning of section 2 (It is not clear to the reader what H1, etc refer to)
- Please report melting temperatures and glass transition temperatures when discussing results in 2.4.
- Some excipients/drugs may undergo cold crystallization during heating of a frozen solution. This can impact the dispersion of ATC in the lyophilized samples. Have the Authors ever tried to investigate this aspect via DSC (cooling the sample to -80°C and then heating to room temperature)?
- Did the Authors use the same freezing protocol for all the samples? Freezing determines the ice crystals morphology and, consequently, the final dried product microstructure. Direct contact with cold surfaces can alter the thermal gradients in the liquid solution undergoing freezing and hence alter the morphology of ice crystals.
English is understandable but needs revision
Reviewer 2 Report
Comments and Suggestions for Authors
Appreciations go to the authors for this interesting read. The rationale for proposing ATC as an option for treatment of oral candidiasis is well-justified: the study explored its effectiveness as an antifungal agent in a novel delivery system, while attempting to combat antimicrobial resistance.
The comparative evaluation of hydrated and freeze-dried forms of the delivery systems is also commendable: lyophilizates showing superior drug release profiles addresses the common solubility issues related to ATC.
It is, however, interesting to know why the authors chose these specific polymer combinations over others since this was not elaborated upon in the introduction/discussion. What is the justification for not using alternative polysaccharides instead of k-carrageenan, xanthan gum etc?
The study reports cytotoxic effects of undiluted hydrogels on HUVEC cells. Could the authors clarify the possibility of the observed antimicrobial effect being associated with unreacted cross-linking agents or byproducts including remnants of DMSO and lactic acid since it was not stated that dialysis was performed to remove all unwanted residues.
Could the authors also please explain why ATC-loaded PECs were not compared with conventional antifungal treatments for cytotoxicity effects? and effectivity in general? This is important for clinical relevance of the study findings.
What are the possible effects of the formulations on beneficial oral microbiota?
Comments on the Quality of English Language
The English quality is generally strong and does not impede comprehension of the paper. However, a few errors require correction, and a thorough review would be beneficial.
See line 121: introduction instead of implementation.
See line 60: remove "being" and "the".
See line 66: remove "the".
See line 70: 'Chitosan,"
See line 78: "alongside" instead of "alongside with".
Reviewer 3 Report
Comments and Suggestions for Authors
The topic of this research is to expand the antimicrobial topical indication area of atorvastatin calcium as a lipid level regulator. To this end, three buccal hydrogel lyophilizates with different formulations were developed. These formulations have been investigated in detail according to the drug formulation assay (in vitro leaching, cytotoxicity, mucoadhesivity and antimicrobial activity measurements, FTIR, SEM, thermal characterization, swelling capacity,).
Basically, the article is well written, the studies are well designed and provide adequate evidence for the formulation.
Basically, the article is well written, the studies are well designed and provide adequate evidence for the formulation.
The figures are also good and informative.
However, I suggest that the novelty of the topic should be better emphasised in the last part of the introduction (line 106- line 110). Why is the new indication of atorvastatin significant, how much better antimicrobial activity does this drug have than the currently marketed ones?
Why is the polyelectrolyte lyophilized hydrogel better than other hydrogels?
